

# Deep learning based respiratory sound analysis for detection of chronic obstructive pulmonary disease

Arpan Srivastava[1], Sonakshi Jain[1], Ryan Miranda[1], Shruti Patil[2], Sharnil Pandya[2] and Ketan Kotecha[2]

[1] CS&IT Dept, Symbiosis Insitute of Technology, Symbiosis International (Deemed University), Pune, Maharastra, India
[2] Symbiosis Centre for Applied Artificial Intelligence, Symbiosis International (Deemed University), Pune, Maharastra, India

Corresponding author
Shruti Patil,
shruti.patil@sitpune.edu.in

## ABSTRACT

In recent times, technologies such as machine learning and deep learning have played a vital role in providing assistive solutions to a medical domain's challenges. They also improve predictive accuracy for early and timely disease detection using medical imaging and audio analysis. Due to the scarcity of trained human resources, medical practitioners are welcoming such technology assistance as it provides a helping hand to them in coping with more patients. Apart from critical health diseases such as cancer and diabetes, the impact of respiratory diseases is also gradually on the rise and is becoming life-threatening for society. The early diagnosis and immediate treatment are crucial in respiratory diseases, and hence the audio of the respiratory sounds is proving very beneficial along with chest X-rays. The presented research work aims to apply Convolutional Neural Network based deep learning methodologies to assist medical experts by providing a detailed and rigorous analysis of the medical respiratory audio data for Chronic Obstructive Pulmonary detection. In the conducted experiments, we have used a Librosa machine learning library features such as MFCC, Mel-Spectrogram, Chroma, Chroma (Constant-Q) and Chroma CENS. The presented system could also interpret the severity of the disease identified, such as mild, moderate, or acute. The investigation results validate the success of the proposed deep learning approach. The system classification accuracy has been enhanced to an ICBHI score of 93%. Furthermore, in the conducted experiments, we have applied K-fold Cross-Validation with ten splits to optimize the performance of the presented deep learning approach.

## INTRODUCTION

The healthcare sector is an independent and one of the most critical sectors compared to various other industries. It is one of the most vital and essential sectors where people expect the highest levels of diagnosis, treatment, and high quality of services in accordance with the money they spend. Various medical equipment's images and audio could have multiple limitations due to their subjectivity, clarity, and complexity. They may be subjected to extensive variations due to a different analysis by different

interpreters/doctors (*Rocha et al., 2019*). Not too long ago, we relied purely on human intelligence, ability and skillset to interpret and understand the enormous amount of medical data generated by various equipment and machinery types. Chronic Obstructive Pulmonary Disease (COPD) is a disease name used as an umbrella of a broad group of lung diseases that block the airflow in and out of the lungs due to narrow air passages, making it difficult to breathe. The lungs are unable to get enough oxygen and give out unwanted carbon dioxide. Emphysema and chronic bronchitis are the two crucial conditions contributing to COPD (*Weese & Lorenz, 2016*). These two conditions usually occur together and can vary in severity among COPD individuals. Bronchitis leads to inflamed and narrowed airways (*Khatri & Tamil, 2018*). Commonly known causes of COPD include tobacco smoking, genetic disorder (alpha-1-antitrypsin deficiency), air pollution, etc. Early or timely detection of COPD is still an area of development which has not achieved 100% accuracy.

Traditionally used techniques by medical practitioners currently are (*Liu et al., 2017*):

1. Lung (pulmonary) function tests: in these tests, we check if the patient's lungs are functioning well and deliver enough oxygen when the person breathes. The most common test for this is called Spirometry. In Spirometry, the person breathes into a device called Spirometer, and that device measures the amount of air the person exhaled.
2. Chest X-ray: the Chest X-Ray can reveal several lung diseases, and especially Emphysema. Emphysema is a major cause of COPD.
3. CT scan: CT Scan is not commonly used but is used in extreme cases where the detection with typical methods is impossible. A CT scan can show if the patient requires any sort of surgery for COPD.
4. Arterial blood gas analysis: this analysis checks the amount of oxygen that increases in the blood when a person breathes.

Limitations of currently used techniques:

1. Spirometry cannot be performed on patients with underlying heart problems or those who have recently undergone heart surgery.
2. Breathlessness, nausea, and dizziness are some symptoms commonly seen after the tests.
3. CT Scans and X-Rays both risk the life of a patient by exposing the patients to radiation.

With so many limitations and the advent of artificial intelligence, various machine learning and deep learning algorithms were examined, studied, and applied to see if they could help provide timely and more accurate results while aiding the doctors in detecting COPD. Detection of COPD using an appropriate algorithm is possible either by images or by the respiratory organ's internal audio.

## RELATED WORK

The related work section describes automation methods and physical systems to ease the detection of respiratory diseases. Physical devices like the Breath Monitoring System (*Radogna et al., 2019*) propose a smart breath analysis device that can be used to detect

COPD. A classifier based on combination of ANN and backpropagation based Multi-Layer Perceptron algorithm to predict patients respiratory audio crucial event with certain respiratory diseases, predominantly asthma and COPD, was examined (*Khatri & Tamil, 2018*). Because ANN is a nonlinear method, its output is better than commonly used classification or regression techniques. Further analysis can be performed along similar lines by refining the classifier's parameters or using specific machine learning techniques such as deep learning. The precision and recall were 77.1% and 78.0% for the peak event level, respectively, and those for the non-peak events were 83.9% and 83.2%. The average machine performance is 81.0%. An articulate system for chronic illness was proposed that offers an integrated platform for successful diagnosis and real-time appraisal of patients' health status with COPD disease (*Bellos et al., 2014*). A machine/device is being put together to observe a patient's condition in real-time. A hybridized classifier was developed on a personalalized digital assistant consisting of a ml algorithms such as SVM, random forest, and a predicate-based approach to include a more complex classification scheme to classify a COPD series early and in real-time. The classification quality obtained was calculated at 94%.

A computer-based approach to automatically interpret stethoscopic recorded respiratory sounds, which has several possible use cases such as telemedicine and self-screening, has also been proposed (*Liu et al., 2017*). One custom-built test tool collects three forms of respiratory sounds from 60 patients. A deep model of Convolutional Neural Networks that consisted of 6 convolution layers, three max-pooling layers, and three wholly linked layers are deployed ahead. Via time-frequency transformation, 60 bands of Log-scaled Mel-Frequency Spectral features were collected framewise which were present in the dataset and segmented as model inputs in a size of 23 consecutive frames. Finally, the developed model was evaluated with a new dataset of 12 subjects measured in precision and recall to 5 respiratory physicians' mean results.

A simple and cost-effective digital stethoscope to record respiratory sounds on a monitor, which can be used on any unit, was proposed by *Aykanat et al. (2017)*, using which 17,930 lung sounds were recorded from 1,630 subjects. The study used two forms of machine learning algorithms: Mel frequency cepstral coefficient features in the Convolutional Neural Network (CNN) and SVM along with spectrogram images. While the usage of MFCC functionality for an SVM algorithm is a widely agreed approach for audio classification, the patrons used its tests to assess the CNN algorithm performance. With each CNN and SVM algorithm, four data sets were prepared for classification of respiratory audio: safe versus unhealthy; rale, rhonchus, and standard classification of speech; singular classification of respiratory speech form; and classification of the audio form of all sound forms. Experimental accuracy findings were 86% CNN, 86% SVM, 76% CNN, 75% SVM, 80% CNN, 80% SVM and 62% CNN, 62% SVM respectively. Consequently, it was found that spectrogram picture classification works well with both the CNN and the SVM algorithms. Given a vast proportion of input, CNN and SVM algorithms can correctly identify and pre-determine COPD through respiratory audio. In research, for the classification of spirometry results, ANFIS, MANFIS, and CANFIS

models with different membership functions were employed (*Asaithambi, Manoharan & Subramanian, 2012*). ANFIS algorithm achieves better recognition accuracy compared to the neural network approach previously employed. This may be because ANFIS incorporates neural networks' learning capacities and the fuzzy inference system's logic capacities. CANFIS achieves 97.5% higher classification accuracy relative to the standard ANFIS and MANFIS, as observed. *Chamberlain et al. (2016)*, focused on wheezes and crackles (two most frequent lung sounds), and their algorithm achieved ROC curves with AUCs of 0.86 and 0.74 for wheezes and crackles, respectively. Another research was carried out in which the COPD disease dataset consisted of 155 samples which were belonging to two distinct categories , classified into Class 1: COPD containing 55 samples and Class 2: Standard containing 100 samples (*Er & Temurtas, 2008*).

Results achieved with an two hidden layer based MLNN (95.33% accuracy) were higher as compared to single hidden layer based MLNN system. A study conducted to categorize data samples used a feed-forward NN architecture with a secret layer running on the log sigmoid transfer feature (*Asaithambi, Manoharan & Subramanian, 2012*). This network had a range of plain, layer-organized neuron-like processing units. The network testing was carried out with the error propagation backward. Neural networks that operate radial bases are often focused on supervised learning. RBF networks can be used effectively to model nonlinear data, as they can be trained in one-step instead of using a repetitive method similar to Multiple Layer Perceptron. Radial base networks learn quickly. The hidden layer outputs are combined linearly in response to an input vector to form the network answer, which is interpreted with a desired solution to the output layer. The weights are conditioned using a standardized linear system in a controlled manner.

Machine-learning techniques have tremendous potential to aid the early detection of COPD exacerbations (*Fernandez-Granero, Sanchez-Morillo & Leon-Jimenez, 2018*). Prediction of exacerbations may reduce inevitable adverse consequences and minimize the high costs associated with patients with COPD. Acute exacerbations are one of the critical factors of declining qualities of a healthy life and accord to sustained obstructive pulmonary disease (COPD) patients being hospitalized. The established model could detect early searing COPD exacerbations 4.4 days before its initiation. For automatic symptom-based exacerbation predictions, a decision tree forest classifier was trained and validated with recorded data.Deep learning based algorithms such as CNN and LSTMS are getting lots of attention in real life critical learning sytems (*Bai et al., 2020*).

## THE NECESSITY OF A PROPOSED SYSTEM

With large datasets of respiratory audio sounds and spectrogram images, we can train deep neural networks without providing lesion-based features to identify COPD condition in patients who are having increased sensitivity and specificity. The main advantage of using this computerized COPD detection system is uniformity in the model outcomes (as a model predicts the same values everytime on a specific image), high sensitivity, dynamic results generation and high specificity. Furthermore, as an algorithm may have several operational points, the responsiveness and precision is possible to tailore to fit different clinical conditions criteria, such as high sensitivity for a screening system.

The conventional methods of COPD diagnosis were detection via a Spirometry test or an AI-based system, which needed images to be fed as input to detect the diseases. In case of any respiratory distress situation like a heart attack or asthma attack, reaching the hospital and making the initial diagnosis via chest scan or x-ray is time-consuming, expensive, and life-threatening. Also, the automated AI systems for image-based detection require training the model on vast numbers of high-quality HD images of x-ray, which is challenging to get each time. Instead, there is a requirement of a more straightforward and less resource-intensive system, which can help medical healthcare providers quickly make the initial diagnosis.

The system that we have proposed is a system that detects COPD based on respiratory sounds. The sound produced by the body's internal organs is very different in case of a heart attack, asthma, COPD, etc. Automated detecting such sounds to classify if a person is susceptible to COPD is a too time-saving, self-alarming method for both the patient and the doctor. The doctors can use the system for confirmed detection of COPD. In contrast, this system's future scope involves integration with smart devices and mics to record people's sounds routinely and thus predict the possibility of a case of COPD.

## Research Gaps in the Existing Literature

Difficulties faced during detection of COPD through audio samples:

1. The previous analysis methods, especially the non-CNNs based neural networks uses highly complex analysis networks, which are very resource-intensive. This implies it requires a high-end computational ability that can incur many infrastructure costs. If there is no investment in the infrastructure, the disease's training and prediction can take a very long time. Existing methods like manual diagnosis by a doctor also takes a long time and several visits to the hospital for concluding if a patient has COPD or not.
2. The number of respiratory audio samples in many cases is quite unbalanced in terms of the diseases. There is always a need to balance the dataset since any network trained on unbalanced data is useful in predicting the disease, which had the highest number of samples.
3. There is usually a lot of noise in the respiratory audio samples. It has to be taken care of in many approaches. Table 1 showcases a summary of the existing research work that has been carried out in the recent past for COPD disease prediction.

## Key aspects of successful deep learning models

1. Selection of Dataset: collecting and managing a clean dataset is highly crucial as the entire model is based on it. It is essential that the dataset used for training be preprocessed with no discrepancies.
2. Selection of Algorithm: it is essential to understand the objective of the study. Various algorithms can be tested to identify which ones yield results closer to achieving the goal.
3. Feature extraction techniques: this is also a vital task in building effective models. It proves to be useful when good model accuracy is required along with optimized feature selection which also helps in generation of redundant data during each data analysis cycle.

**Table 1 Existing work.**

| Year | Author | Purpose | Techniques used | Accuracy | Features | Issues |
|---|---|---|---|---|---|---|
| 2020 | Du et al. (2020) | To detect COPD using snapshots of 3D lung airway tree and with the help of deep CNN | Classification by colorful snapshot | 88.2% | Features from Spirometry Tests | Small dataset |
| | | | Classification by gray snapshot | 88.6% | | Includes only COPD and healthy control patients |
| | | | Classification by binary snapshot | 86.4% | | Utilizes the information of the airway, but does not include lung parenchyma, patterns of LAA, air trapping, pulmonary blood vessels, and medical records in the deep CNN model. |
| 2020 | Ahmed et al. (2020) | Classification of COPD using CT images and a 3D Convolutional Neural Network | Without Transfer Learning | Validation: 68.5%, Test accuracy: 58.8% | CT images from two datasets consisting of inspiration scans using soft kernel | Validating the model with a detailed comparison with other methods on bigger and balanced data sets, ideally using k-fold cross-validation. |
| | | | With Transfer Learning | Validation: 78.3%, Test accuracy: 70% | | |
| 2008 | Manoharan, Veezhinathan & Ramakrishnan (2008) | Comparisons of two artificial neural networks based on spirometric pulmonary function tests data | Back Propagation | 96% | Individual Features from spirometry test. | The correctness can be further enhanced by adding more input feature values (spirometric) and a huge database for training. |
| | | | Radial basis function Neural Network | 100% | | |
| 2008 | Er & Temurtas (2008) | Detection of COPD via utilizing two MultiLayer NN structures, one having a two hidden layer and other having one single hidden layer. | MLNN with: | | Individual Features were used. | It was noticed that the neural network with 2 hidden layers was more preferable than with 1 hidden layer. However, they have not checked for more than 2 hidden layers that could yield even better results. |
| | | | one hidden Layer + Backpropagation | 93.14% | | |
| | | | one hidden Layer + Levenberg–Marquardt algorithm | 94.46% | | |
| | | | two hidden layers + Backpropagation | 95.43% | | |
| | | | two hidden layers + Levenberg–Marquardt algorithm | 96.08% | | |
| 2012 | Amaral et al. (2012) | To identify chronic obstructive pulmonary disease using applied machine learning algorithms and forced oscillation measurements | Worked on the selection of features for various classification algorithms such as linear bayesian , KNN, decision trees, ANN, and SVM | KNN, SVM and ANN were found to be the best with Se > 87%, Sp > 94%, and AUC > 0.95. | | Does not classify the severity of the disease whether mild, moderate or severe. |

| Year | Author | Purpose | Techniques used | Accuracy | Features | Issues |
|---|---|---|---|---|---|---|
| 2016 | Chamberlain et al. (2016) | Classification of lung sounds using a deep learning algorithm (semi-supervised) that emphasizes wheezes and crackles. | Denoising Auto Encoder | 86% Wheeze | 30-second audio recordings from 11 different chest locations were used from the patients. | None |
| | | | | 74% Crackle | | |
| | | | | Two SVMS have been trained in Denoising Autoencoder to individually identify wheezes and identify crackles respectively. | | |
| 2018 | Altan et al. (2018) | Analysis of 3D-space quantization that originates with successive three data points in the signal. | Deep Belief Networks | Accuracy: 95.84%, Sensitivity: 93.34% and Specificity: 93.65% | 120 lung sounds from smokers (12 channels x 5 subjects) and COPD patients (12 channels x 5 subjects) were used. | |
| 2012 | Asaithambi, Manoharan & Subramanian (2012) | To classify respiratory abnormalities with the help of an Adaptive Fuzzy Inference System | **Adaptive Neuro-Fuzzy Inference System (ANFIS)** | | No specific feature selected | Severity classification of respiratory abnormalities not done |
| | | | Triangular | 88.17% | | |
| | | | Trapezoidal | 82.22% | | |
| | | | Gaussian | 85.56% | | |
| | | | Gbell | 88.89% | | |
| | | | **Multiple ANFIS** | | | |
| | | | Triangular | 93.33% | | |
| | | | Trapezoidal | 96.67% | | |
| | | | Gaussian | 85.56% | | |
| | | | Gbell | 84.44% | | |
| | | | **Complex valued ANFIS** | | | |
| | | | Triangular | 91.25% | | |
| | | | Trapezoidal | 90.82% | | |
| | | | Gaussian | 97.55% | | |
| | | | Gbell. | 82.50% | | |

| Year | Author | Purpose | Techniques used | Accuracy | Features | Issues |
|------|--------|---------|-----------------|----------|----------|--------|
| 2018 | *Fernandez-Granero, Sanchez-Morillo & Leon-Jimenez (2018)* | Early predictions on symptom-based COPD exacerbations using Artificial Intelligence | Decision Tree Classifier | Accuracy 87.8%, Sensitivity 78.1%<br><br>Specificity 95.9% | Features were extracted using a discrete wavelet transform. The input dataset contained 18 wavelet features. | Not large enough training data and larger sample data needed to get more accurate results to determine the actual accuracy of the model. Longer training time on audio |
| 2018 | *Badnjevic, Gurbeta & Custovic (2018)* | Identification of asthma and chronic obstructive pulmonary disease automatically using an Expert Diagnostic System | Artificial Neural Network + Fuzzy Logic | Copd: 95.17%<br>Healthy: 98.7% | Seven features from symptom questionnaire + four features from the result of SPIR test | Evaluation of the EDS was conducted in a healthcare center in Bosnia and Herzegovina. However, it is recommended to further test in other healthcare establishments in order to further justify the EDS. |
| 2018 | *Altan, Kutlu & Allahwardi (2019)* | Deep Learning for COPD Detection using Lung Sounds | Deep Belief Networks<br><br>Sequential Forward Feature Selection applied to DBN | 70.28%, 67.22%, 73.33% as accuracy, sensitivity and specificity<br><br>90.83%, 94.44%, 87.22% as accuracy, sensitivity and specificity | No specific feature selected | Results suffered from noise presence. Overall also, the accuracy, sensitivity and specificity are less without SFFS. |

# OVERVIEW OF THE PROPOSED SYSTEM

As shown in Fig. 1, the developed application can be described as a series of interconnected processes across multiple layers. The user layer consists of the doctor taking the recording of lung sounds. These recordings are then forwarded to the front-end layer, which is the interface with which the doctor can upload the patients recording into the system. The network layer provides connectivity and access to the back-end layer on the cloud. Finally, the back-end layer consists of a feature extraction module and a classifier module to analyze and provide results back to the front-end interface.

The classification process is broken into several stages. The audio sample is first recorded and passed to the preprocessing module to transmute the data. The data is normalized, and features from this altered data are extracted and given to the augmentation module. Upon completion of the augmentation process, it is passed to the convolutional neural network classified into either of the two categories: COPD and non-COPD. This system can be used as an aid to the doctors who are diagonosing respiratory diseases in patients. In case of severe COPD patients, it will help in early by capturing the sound of breathing instead of waiting for the chest x-ray reports.

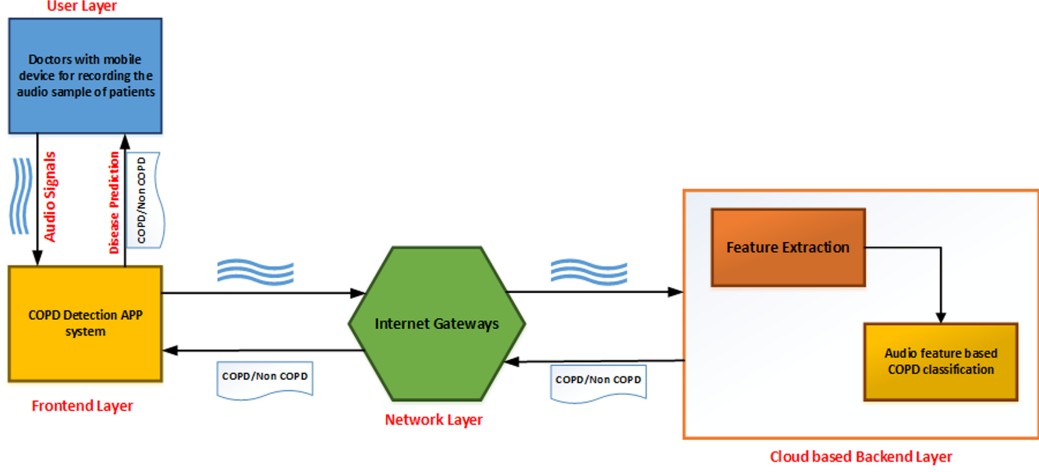

**Figure 1 Topology design of the proposed system.**

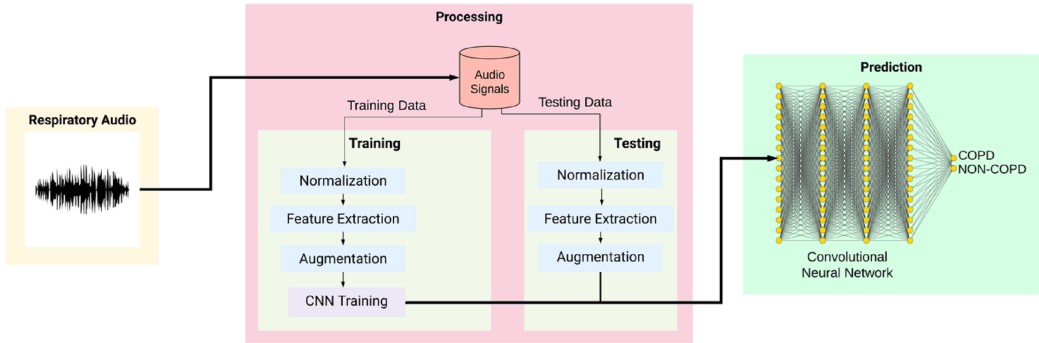

**Figure 2 Workflow of the proposed system.**

## Methodology

Figure 2 shows the overall training and testing methodology and is explained as given below:

Data Acquisition: the lung sounds that are offered as input was recorded from normal as well as abnormal male and female patients with various kinds of respiratory dysfunctions such as: COPD, Asthma, lower and upper respiratory tract infection (LRTI, URTI). Two research teams from Portugal and Greece created a database of respiratory sounds from where 126 input recordings has been taken (*ICBHI, 2017* Challenge). The data samples include both respiratory sounds of healthy individuals as well as of patients who were having the repiratory ailments. The patients span from all age groups, including young children, adults, and senior citizens. The dataset consists of a total of 5.5 h of recordings containing 6898 respiratory cycles, of which 1,864 contain crackles, 886 contain wheezes and 506 contain both crackles and wheezes, in 920 annotated audio samples from 126 subjects. The cycles were annotated by respiratory experts as including crackles, wheezes, a combination of them, or no adventitious respiratory sounds. The recordings were collected using heterogeneous equipment and

| Table 2 Augmentation methods. | | |
|---|---|---|
| No. | Augmentation technique | Parameters |
| 1 | Loudness augmentation | 0.6–1.4% of loudness |
| 2 | Mask augmentation | Random areas masked |
| 3 | Shift augmentation | Shifts random segments in random(left/right) directions |
| 4 | Speed augmentation | Between 0.5× and 2× speed |

their duration ranged from 10 s to 90 s. The chest locations from which the recordings were acquired is also provided. Noise levels in some respiration cycles is high, which simulate real life conditions.

Data Preprocessing: the dataset contained a lot of irregularities and unstructured data. To normalize the data, we trimmed/padded the audio files to a length of 20 seconds using a Python library Librosa (*Librosa, 2020*).

Feature Extraction: for the feature extraction, we calculated five features. The features were Mel-Frequency Cepstral Coefficients (mfcc), melspectrogram (Mel-Spectrogram), Chromagram calculated from the waveform/power spectrogram (chroma_stft), Constant-Q Chromagram (chroma_cqt) and chroma_cens (Chroma Energy Normalized Variant (CENS)). MFCCs are coefficients that collectively make up an mel-frequency cepstrum (MFC). An MFC is a representation of the short-term power spectrum of a sound, based on a linear cosine transform of a log power spectrogram on a non-linear mel scale of frequency. These features represent phonemes (which are the distinct units of sound) as the shape of the vocal tract (which is responsible for sound generation) is manifest in them. This makes MFCC a great feature to consider for respiratory audio analysis. In order to obtain the Mel-Spectrogram, we take samples of air pressure over time, map it from the time domain to the frequency domain using the fast Fourier Transform and we convert the freuqnecy to a mel scale and the color dimension to the amplitude. It represents short-term power spectrum of a sound. Chroma-based features (like the ones we mentioned above) are also referred to as "pitch class profiles", are a powerful set of features for analysing music whose pitches can be categorized. Since the respiratory sounds also vary quite distinctly in pitch, Chroma makes it a great feature for our user case. CENS features are robust to dynamics, timbre, and articulation, making these commonly used in audio matching and retrieval applications. We gave each of the features the "n" value (like n_mfcc's in mfcc and n_chroma bins in chroma features) as 40 to maintain consistency across the features.

Augmentation: we used different audio augmentation methods on the samples to increase the number of non-COPD samples since the number of COPD samples was almost four times the number of non-COPD samples. We applied the following techniques for the audio augmentation, as shown in Table 2.

CNN Structure: our model is a CNN developed using Keras and a Tensorflow back-end. It is a sequential model comprising of Input Layer, Convolution 2D layers, DropOut Layers, MaxPooling2D Layers and a Dense Layer. Feature detection is the main aim of a

**Table 3 CNN architecture.**

| Network architecture | Output |
|---|---|
| Input layer (Image dimensions: $40 \times 862 \times 1$) | |
| Conv2D (16F, K2, ReLu) - MaxPooling2D (P2) - Dr 0.2 | $19 \times 430 \times 16$ |
| Conv2D (32F, K2, ReLu) - MaxPooling2D (P2) - Dr 0.2 | $9 \times 214 \times 32$ |
| Conv2D (64F, K2, ReLu) - MaxPooling2D (P2) - Dr 0.2 | $4 \times 106 \times 64$ |
| Conv2D (128F, K2, ReLu) - MaxPooling2D (P2) - Dr 0.2 | $1 \times 52 \times 128$ |
| GlobalAverage2D | 128 |
| Dense | 2 |

convolution layers. It works by moving a filter window over the input and calculating the multiplication of matrices which gets stored as a feature map. This process of feature map generation is known as a convolution. Each convolutional layer has an associated pooling layer of MaxPooling2D, with the final layer having GlobalAveragePooling2D type. The task of reducing the model dimentionability is done by the pooling layer which is achieved by optimizing the parameters and leads to less computation requirements. This reduces the training time and chances of overfitting. For each window, the maximum size and average window size is taken by Maxpooling and the Global Average Pooling type for feeding into our dense output layer. The count of possible classification is decided based on the two nodes (num_labels of the output layer). The activation of the output layer is softmax. Softmax makes the output number as high as one such that the output can be viewed as a probability.

The quantity of nodes in each layer is specified by the filter parameter. In a progressive manner, the size of the individual layer increase from 16, 32, 64 to 128, while the kernel size parameter defines the kernel window size, which is two resulting in a $2 \times 2$ filter matrix. As shown in Table 3, an input shape will be provided to the first layer. The values of input shape would be of (40, 862, 1) representing number of MFCC's as 40, number of frames as 862 considering padding into account, and the mono audio structure showed by 1. The features then go through the Convolution2D layer (16 filters, kernel size: 2, relu) which passes on the data to the MaxPooling2D layer of (Pooling Size: 2). After this we set a 20%dropout rate to avoid overfitting the data. After the dropout, the data goes to a Convolution layer (32 filters, kernel size:2, relu), which again passes on the data to a MaxPooling2D layers of (Pooling size: 2). Once again after a 20% dropout, there is a Convolution2D layer (64 filters, kernel size: 2, relu), MaxPooling2D layer (Pooling size: 2) and a dropout of 20%. The data then passes through another Convolution2D layer (32 filters, kernel size:2, relu), MaxPooling2D layer (Pooling size: 2) and a dropout layer (20%). ReLu is the activation function used for generation of activation map from convolutional layer. Then finally we pass the remaining data to the GlobalAveragePooling2D layer (128) to flatten the output and then finally pass it to a dense layer to classify it into 2 outputs, namely COPD and non-COPD.

It is observed that, the audio inputs generally have higher amount of noise as compared to the image inputs. We have used the Adam optimization algorithm for optimizing

**Table 4 Comparative analysis of various optimizers on benchmark datasets** *Wierenga (2020)*.

| Optimizers | Training accuracy (%) | Training loss | Validation accuracy (%) | Validation loss |
|---|---|---|---|---|
| MNIST | | | | |
| SGD | 89.54 | 0.3798 | 90.27 | 0.3657 |
| AdaGrad | 91.06 | 0.3228 | 91.45 | 0.3167 |
| RMSProp | 99.37 | 0.0248 | 97.67 | 0.0813 |
| Adam | 99.56 | 0.0210 | 97.60 | 0.0790 |
| CIFAR10 | | | | |
| SGD | 47.02 | 1.4931 | 46.03 | 1.5064 |
| AdaGrad | 48.56 | 1.4463 | 47.58 | 1.4537 |
| RMSProp | 85.65 | 0.4479 | 76.23 | 0.7176 |
| Adam | 92.46 | 0.2841 | 77.97 | 0.6477 |
| IMDB Reviews | | | | |
| SGD | 90.77 | 0.2766 | 87.43 | 0.3334 |
| AdaGrad | 90.75 | 0.2770 | 87.39 | 0.3341 |
| RMSProp | 90.79 | 0.2771 | 87.45 | 0.3338 |
| Adam | 90.86 | 0.2755 | 87.55 | 0.3326 |

our model as it is computationally efficient, requires less memory storage and gives a better performance for noisy input values. Adam is a combination of stochastic gradient descent and RMSprop algorithm, which provides an improved network weight optimization logic that in turn also makes the process of hyper parameter tuning more efficient (*Kingma & Ba, 2017*). As compared to other existing optimers such as RMSprop, SGD , ADAGrad, NAdam this algorithm provides a faster convergence along with an optimised performance parameters. A comparative analysis of various optimizers on three benchmarked database was refrerred and shown in the below given Table 4:

The model then makes its prediction based on which option has the highest likelihood. The provided audio sample is split into two sets of data, training data and testing data. Before training the model, the training data is run through a series of processes like Normalization, Feature Extraction, and Augmentation. The model then stores the training data with its modified features. The testing data, likewise, is run through a similar series of processes. It uses the trained model created for CNN classification and prediction of COPD. A spectrogram is useful in describing the signal in terms of time, frequency, and magnitude. To define, a spectrogram is a 2D Representation of audio with time and magnitude being the two dimensions, with a third dimension depicted by colors. Since respiratory diseases are musical anomalies identified by proper auscultation techniques, their presence also produces a different spectrogram. Our extracted features can be divided into three main categories, namely Mel-Spectrograms, MFCC and Chromagram.

The vocal tract's shape manifests itself in the short-time power spectrum envelope, and the MFCC accurately represents this envelope. MFCC's have been widely used in audio classification since they were introduced. Chroma features, also known as pitch-class features, are a powerful representation for an audio sample. The normal Chroma variant (chroma_stft) typically indicates how much energy of each pitch class is present in the

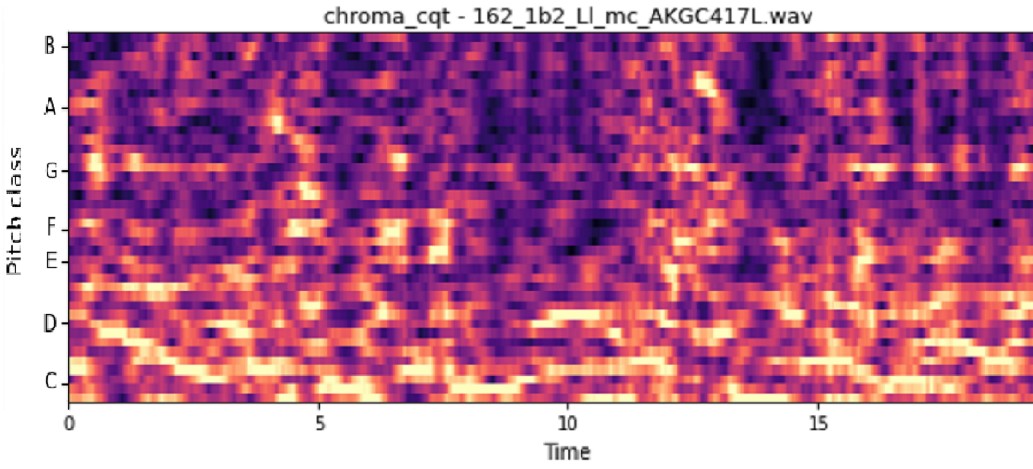

**Figure 3 Chroma constant-Q time.** *X*-axis: pitch class; *y*-axis: time.

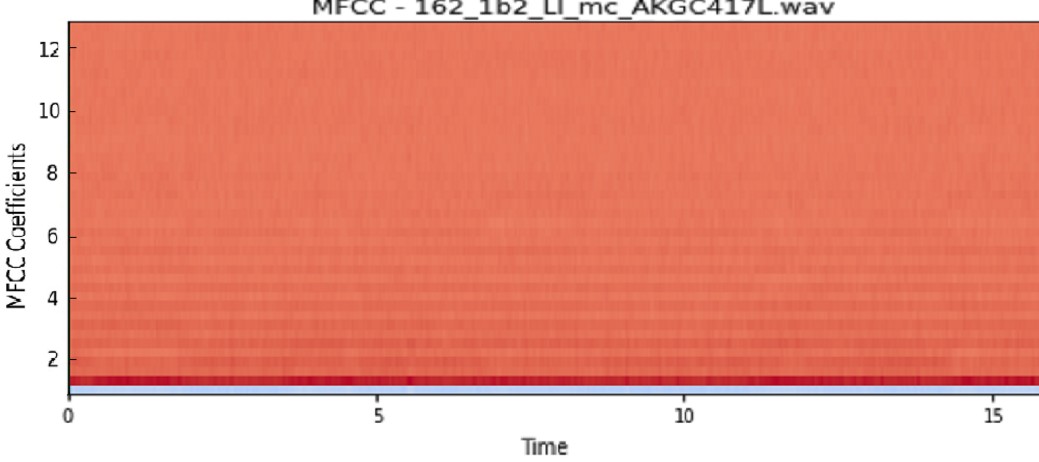

**Figure 4 MFCC representation.** *X*-axis: MFCC coefficients; *y*-axis: time.

signal. We have also used the Chromagram with a "Constant Q" transform (chroma_cqt). The constant-Q transform transforms a data series from a time domain to a frequency domain. The other variant, Chroma Energy Normalized Statistics (CENS) (chroma_cens), is based on the idea that taking statistics over large windows smooths local deviations in different audio characteristics (*Librosa, 2019*). Figures 3–7 shows the spectral representations that were produced and considered for the analysis.

Device Configuration: all this was run on a machine with the configuration as listed below:

- Processor: Intel(R) Core(TM) i7-7700HQ CPU @ 2.80GHz
- GPU: Nvidia K80s, (24GB DDR5 Type)
- RAM: 36GB
- HDD: 250GB SSD, max sequential read speed: upto 550 MB/s, max sequential write speed: upto 520 MB/s

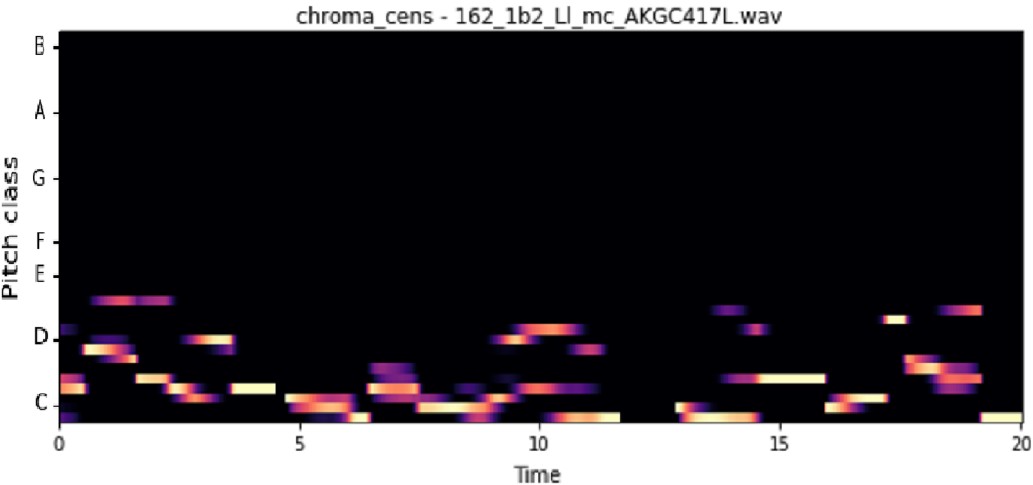

**Figure 5 Chroma CENS representation.** *X*-axis: pitch class; *y*-axis: time.

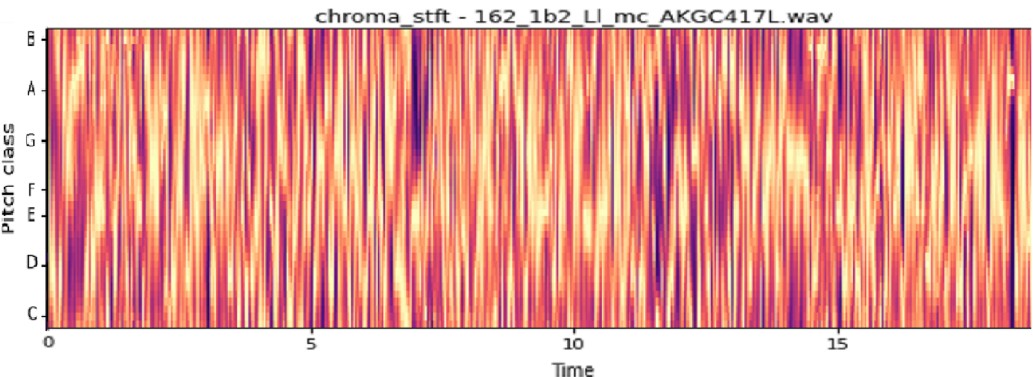

**Figure 6 Chroma STFT representation.** *X*-axis: pitch class; *y*-axis: time.

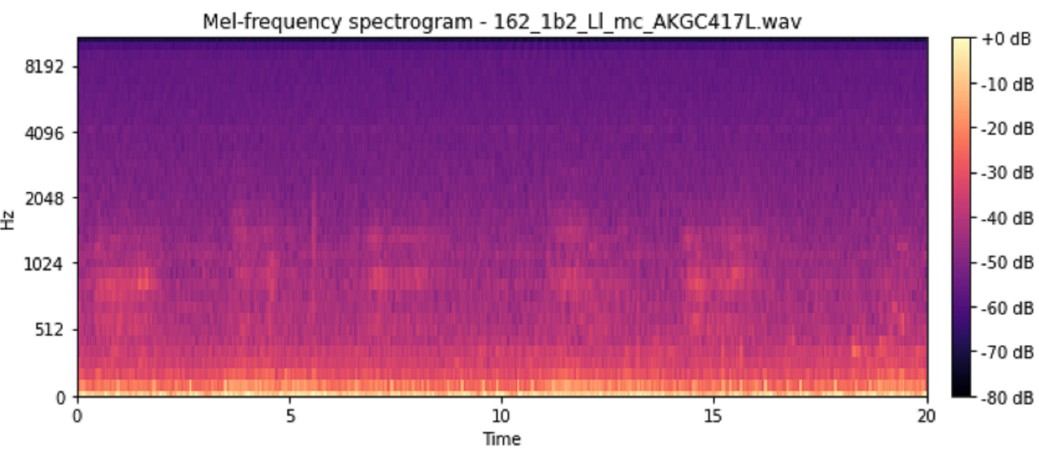

**Figure 7 Mel frequency spectrogram.**

1["

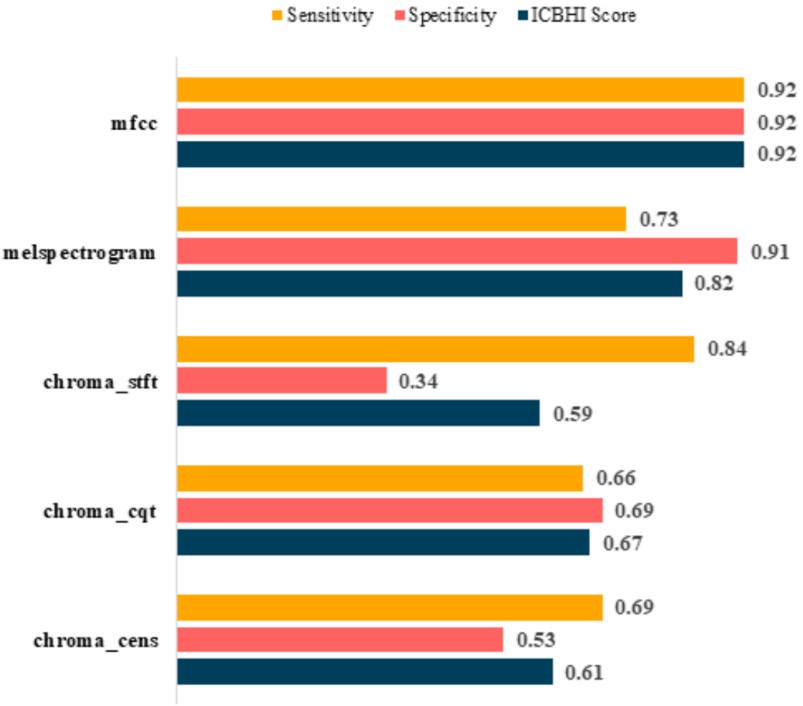

Figure 9 Post-augmentation metrics graph.

Table 6 Post-augmentation metrics.

| Feature | Sensitivity | Specificity | ICBHI score |
|---|---|---|---|
| mfcc | 0.92 | 0.92 | 0.92 |
| melspectrogram | 0.73 | 0.91 | 0.82 |
| chroma_stft | 0.84 | 0.34 | 0.59 |
| chroma_cqt | 0.66 | 0.69 | 0.67 |
| chroma_cens | 0.69 | 0.53 | 0.61 |

Post-augmentation

It is evident that the values after augmentation have improved significantly. The Sensitivity and Specificity of mfcc were consistent, this time both being 0.92. None other feature worked well after augmentation. Table 6 shows the post augmentation values for various components.

Area Under Curve (AUC): the best AUC for the features before augmentation was achieved by "mfcc" (0.86). Using Augmentation, AUC for "mfcc" increased to 0.89. All other features performed poorly compared to "mfcc". The actual values are shown in Table 7. The graphical representation of pre and post augmentation performance is shown in Fig. 10.

Receiving Operator Characteristic (ROC): as shown in Fig. 11, the feature "mfcc" had the best ROC Curve before Augmentation. Using augmentation, it offered an increase of

| Table 7 AUC before and after Augmentation. | | |
|---|---|---|
| **Feature** | **AUC** | |
| | **Pre-augmentation** | **Post augmentation** |
| mfcc | 0.86 | 0.89 |
| melspectrogram | 0.62 | 0.77 |
| chroma_stft | 0.59 | 0.62 |
| chroma_cqt | 0.71 | 0.65 |
| chroma_cens | 0.53 | 0.60 |

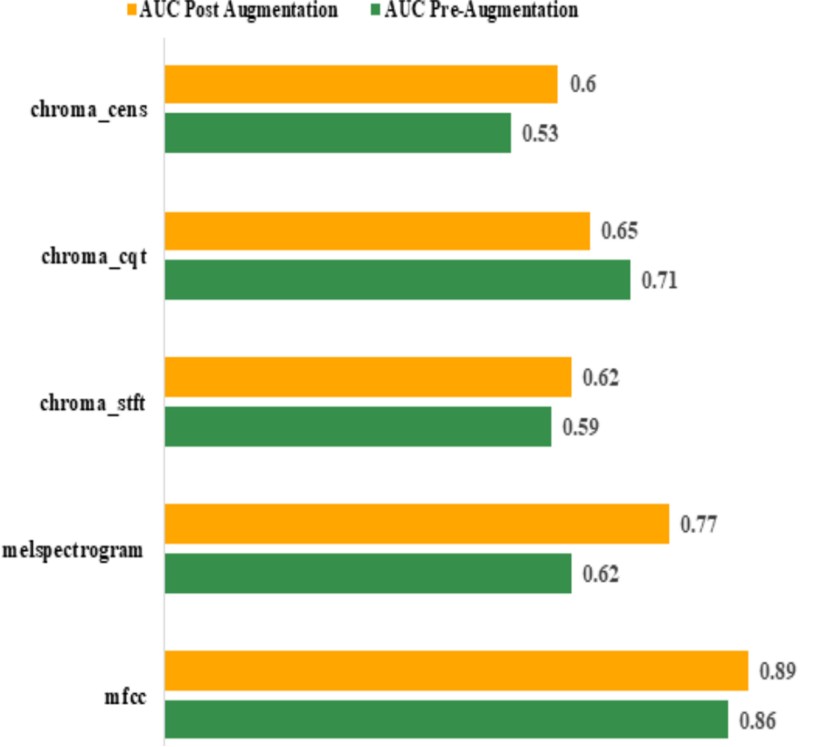

**Figure 10 Pre and post augmentation AUC performance.**

10% in the AUC. We can interpret two significant results after our analysis. We can see that "mfcc" clearly outperformed every other feature in all the metrics. Consequently, we have used the results of "mfcc" to compare against different state-of-the-art methodologies.

When we compare our model (mfcc) with other models, as shown in Fig. 12 and Table 8, we can see that the presented model has a competitive and balanced score in terms of Sensitivity, Specificity. It outperforms the other models' performance, which is presented in terms of the ICBHI Score.

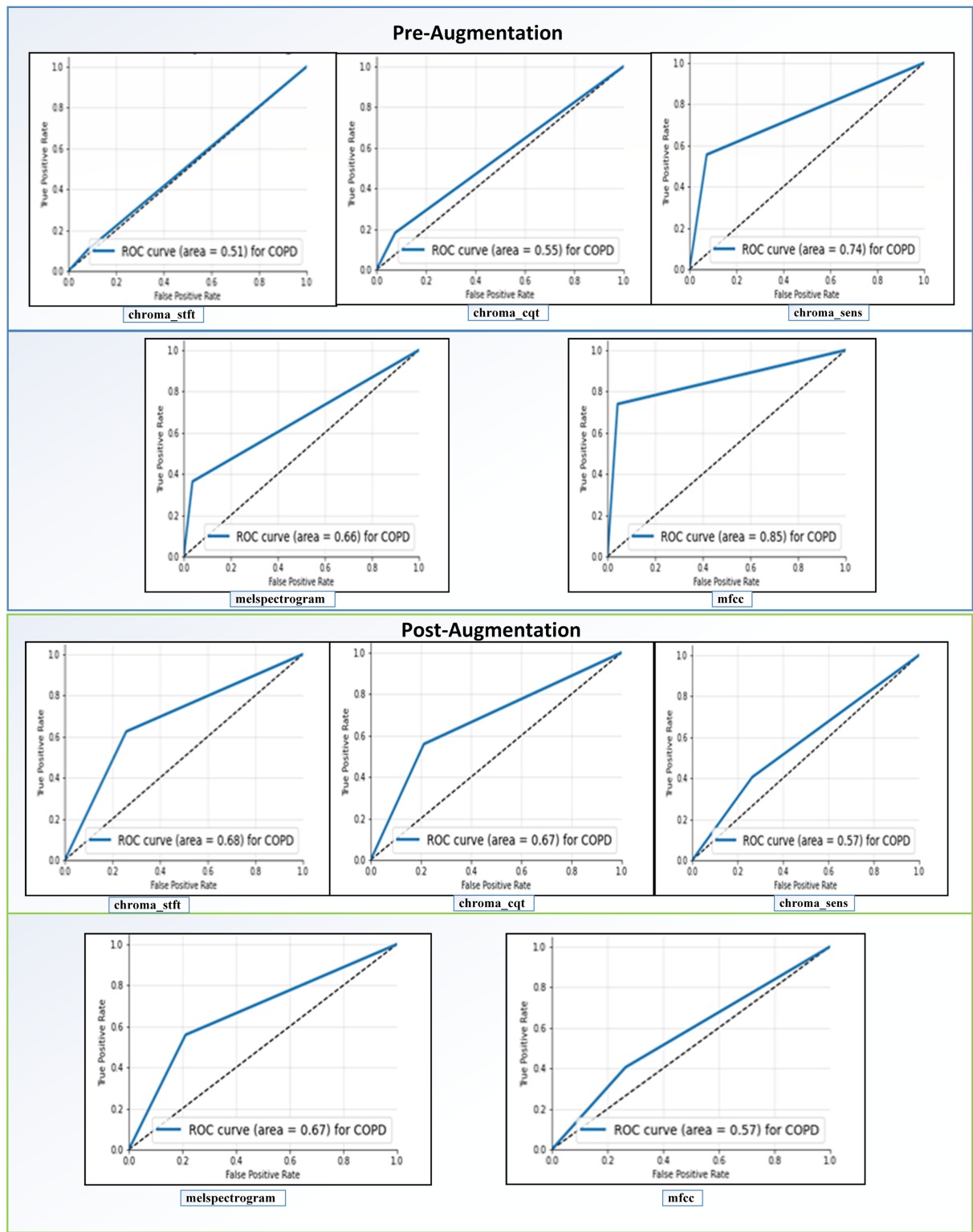

**Figure 11 ROC curves before and after augmentation.**

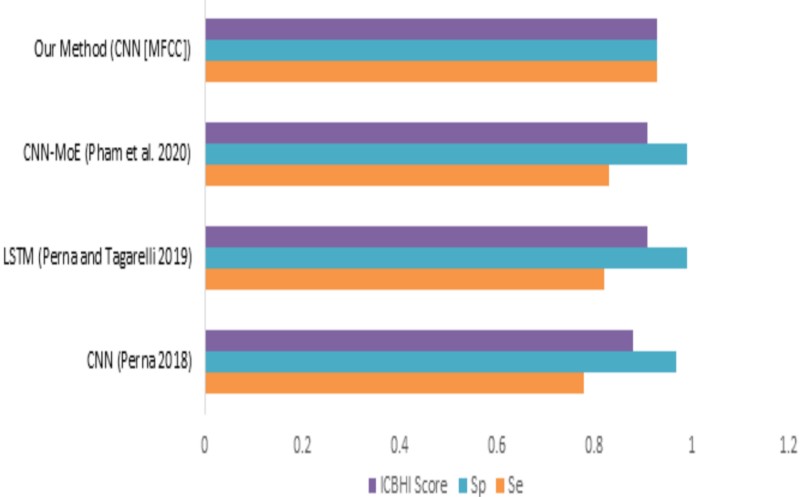

**Figure 12 Comparison of features with existing techniques.**

**Table 8 Comparison of features with existing techniques.**

| Method | Sensitivity | Specificity | ICBHI score |
|---|---|---|---|
| CNN (_Perna, 2018_) | 0.78 | 0.97 | 0.88 |
| LSTM (_Perna & Tagarelli, 2019_) | 0.82 | 0.99 | 0.91 |
| CNN-MoE (_Pham et al., 2020_) | 0.83 | 0.99 | 0.91 |
| Our method (CNN [MFCC]) | 0.93 | 0.93 | 0.93 |

## CONCLUSION

In this study, we have proposed a simple and less resource-intensive CNN-based deep learning assistive model, which can assist medical experts in detecting COPD using respiratory sounds. In the conducted experiments, we have used a Librosa machine learning library features such as MFCC, Mel-Spectrogram, Chroma, Chroma (Constant-Q), and Chroma CENS to perform a detailed analysis of respiratory sounds (Librosa Chroma feature overview, Accessed 20 May 2020) on the (_ICBHI, 2017_) Dataset. Based on the conducted experiments, it has been found that "mfcc" has provided better accuracy in detecting COPD compared to all other Librosa machine learning library features. For the future scope, we can extend its functionalities to aid doctors in detecting various other diseases such as probability of occurrence of heart attack/cardiac arrest based on the heart beats sounds, detection of Asthma based on lung sounds etc. We can also enhance the current system to detect the severity of the disease. We can also apply different data augmentation techniques to improve its performance. Our system can be combined with a Breath Monitoring System (_Radogna et al., 2019_) to make the process of detecting COPD even easier. Also, making this system attack resistant and more privacy preserved would be of utmost importance (_Shaikh & Patil, 2018_; _Patil, Shashank & Deepali, 2020_).

## APPENDIX

List of Used Terminologies

| | | | |
|---|---|---|---|
| CNN | Convolutional Neural Network | CANFIS | Complex-Valued Adaptive Neuro-Fuzzy Inference System |
| COPD | Chronic Obstructive Pulmonary Disease | ROC | Receiving Operator Characteristic |
| ANN | Artificial Neural Network | AUC | Area Under Curve |
| SVM | Support Vector Machines | MLNN | Multi-Layer Neural Network |
| ANFIS | Adaptive Neuro-Fuzzy Inference System | BP | Back Propagation |
| MANFIS | Multiple Adaptive Neuro-Fuzzy Inference System | LM Structure | Levenberg-Marquardt Structure |
| RBF Network | Radial Basis Function Network | KNN | K-Nearest Neighbors |
| ReLU | Rectified Linear Unit | MFCC | Mel-Frequency Cepstral Coefficients |

### Funding

The authors received no funding for this work.

### Competing Interests

The authors declare that they have no competing interests.

### Author Contributions

- Arpan Srivastava conceived and designed the experiments, performed the experiments, analyzed the data, performed the computation work, prepared figures and/or tables, authored or reviewed drafts of the paper, and approved the final draft.
- Sonakshi Jain performed the experiments, performed the computation work, authored or reviewed drafts of the paper, and approved the final draft.
- Ryan Miranda performed the experiments, performed the computation work, prepared figures and/or tables, authored or reviewed drafts of the paper, and approved the final draft.
- Shruti Patil conceived and designed the experiments, analyzed the data, performed the computation work, prepared figures and/or tables, authored or reviewed drafts of the paper, and approved the final draft.
- Sharnil Pandya performed the experiments, analyzed the data, performed the computation work, authored or reviewed drafts of the paper, and approved the final draft.
- Ketan Kotecha analyzed the data, authored or reviewed drafts of the paper, and approved the final draft.

### Data Availability

The code for COPD research is available at GitHub:

https://github.com/ArpanSriv/copd-research.

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
