# Peer review of "Deep learning based respiratory sound analysis for detection of chronic obstructive pulmonary disease"

_PeerJ Computer Science, doi:10.7717/peerj-cs.369_

## Round 0.1 · original submission · Major Revisions

Dear Authors, Reviewers suggest changes to your paper by solving important issues. Please revise it and return to us with revisions for further processing.

Reviewer 1 ·

Basic reporting

This article shows an interesting solution that fits into the current needs of the healthcare system around the World. Authors have created a well performing CNN model recognizing, in a very short time, Chronic Obstructive Pulmonary Disease (COPD). Because of that it could help the doctors, especially in overpopulated areas, to diagnose, with higher accuracy, more patients in the same time, which could potentially lead to faster treatment and less health complications.
The article is well and clearly written and provides all used resources, however some minor changes are required.

Experimental design

After analyzing your code I can see that in your work you have used Adam optimization algorithm, however it is not mentioned in your article. As we know there are many different training algorithms and because of that I have a question:
- Why have you chosen such algorithm? Were there any experiments carried out with other algorithms like NAdam, RMSprop, etc.? And how does they compare with each other (training times, accuracy, ...)

You have mentioned that other systems takes a very long time to train and predict. How does it perform in your case? Is it faster than other methods with similar results? What is the device requirenment to run the network with reasonable predicting times and to train the model?

Validity of the findings

All findings are well documented, clear to read and does not contain any major errors.

Additional comments

This article is well made and contains valuable knowledge. It's easy to read.

Reviewer 2 ·

Basic reporting

Table VII should be corrected: the headers are large, axis descriptions illegible.

Table VIII explain word abbreviations Se Sp

Fig 3-6 no axes description.

Fig 3-fig 11 are large and not clear

Experimental design

More precisely explain and describe the feature extraction methods between input analogic signal, spectrogram and input to your neural network.

More precisely explain how the architecture of convolutional neural networks looks like (convolutional, pooling). Can you show some examples of each operation?

Validity of the findings

Authors shows interesting applied of CNN for detection of chronic obstructive pulmonary disease, but numer of examples form experiments is small, so give more examples for other types of images

Additional comments

Explain where the presented method can be used in practice.

How this method can be improved and developed in the future?

Authors can extend the bibliography to include articles: Bio-inspired voice evaluation mechanism, DecomVQANet: Decomposing visual question answering deep network via tensor decomposition and regression

---

## Round 0.2 · accepted · Accept

Dear Authors the review process is now finished. All Reviewers are positive on your manuscript therefore i am pleased to forward your manuscript with a positive recommendation.

Reviewer 1 ·

Basic reporting

Article is overall well written and contains interesting and revealing content. Authors have used professional English and included accurate literature references. Proofs are clear and does not contain any visible errors.

Experimental design

Authors have replied to all given critique and have given clear responses.

Validity of the findings

All findings are well documented, clear to read and does not contain any visible errors.

Additional comments

This article is well made and contains valuable knowledge. It's easy to read.

Reviewer 2 ·

Basic reporting

Check fig. 3,4,6 axis descriptions are truncated.
I have no more critical comments.

Experimental design

In my opinion it is ok.

Validity of the findings

OK

Additional comments

The authors followed my suggestions so I have no further objections.